# Practical Deep Stereo (PDS): Toward applications-friendly deep stereo matching.

**Stepan Tulyakov**
Space Engineering Center at
École Polytechnique Fédérale de Lausanne
`stepan.tulyakov@epfl.ch`

**Anton Ivanov**
Space Engineering Center at
École Polytechnique Fédérale de Lausanne
`anton.ivanov@epfl.ch`

**Francois Fleuret**
École Polytechnique Fédérale de Lausanne
and Idiap Research Institute
`francois.fleuret@idiap.ch`

## Abstract

End-to-end deep-learning networks recently demonstrated extremely good performance for stereo matching. However, existing networks are difficult to use for practical applications since (1) they are memory-hungry and unable to process even modest-size images, (2) they have to be trained for a given disparity range.

The Practical Deep Stereo (PDS) network that we propose addresses both issues: First, its architecture relies on novel bottleneck modules that drastically reduce the memory footprint in inference, and additional design choices allow to handle greater image size during training. This results in a model that leverages large image context to resolve matching ambiguities. Second, a novel sub-pixel cross-entropy loss combined with a MAP estimator make this network less sensitive to ambiguous matches, and applicable to any disparity range without re-training.

We compare PDS to state-of-the-art methods published over the recent months, and demonstrate its superior performance on FlyingThings3D and KITTI sets.

## 1 Introduction

Stereo matching consists in matching every point from an image taken from one viewpoint to its physically corresponding one in the image taken from another viewpoint. The problem has applications in robotics [22], medical imaging [23], remote sensing [32], virtual reality and 3D graphics and computational photography [37, 1].

Recent developments in the field have been focused on stereo for hard / uncontrolled environments (wide-baseline, low-lighting, complex lighting, blurry, foggy, non-lambertian) [36, 11, 3, 5, 27], usage of high-order priors and cues [9, 8, 14, 17, 34], and data-driven, and in particular, deep neural network based, methods [25, 3, 39, 40, 19, 33, 30, 16, 31, 7, 13, 20, 24, 2, 18, 43]. This work improves on this latter line of research.

The first successes of neural networks for stereo matching were achieved by substitution of hand-crafted similarity measures with deep metrics [3, 39, 40, 19, 33] inside a legacy stereo pipeline for the post-processing (often [21]). Besides deep metrics, neural networks were also used in other subtasks such as predicting a smoothness penalty in a CRF model from a local intensity pattern [30, 16]. In [31] a "global disparity" network smooth the matching cost volume and predicts matching confidences, and in [7] a network detects and fixes incorrect disparities.

Table 1: Number of parameters, inference memory footprint, 3-pixels-error (3PE) and mean-absolute-error on FlyingThings3D ($960 \times 540$ with 192 disparities). DispNetCorr1D [20], CRL [24], iResNet-i2 [18] and LRCR [12] predict disparities as classes and are consequently over-parameterized. GC [13] omits an explicit correlation step, which results in a large memory usage during inference. Our PDS has a small number of parameters and memory footprint, the smallest 3PE and smallest or second smallest MAE, depending on evaluation protocol, and it is the only method able to handle different disparity ranges without re-training. Note, that for our method we report two results. The result outside of brackets is obtained using protocol of PSM [2] method, according to which the errors are calculated only for ground truth pixel with disparity $< 192$. The result in the brackets is calculated according to protocol of CRL [24], DispNetCorr1D [20] and iResNet-i2 [18] methods, according to which the error is calculated only for images where less than 25% of pixels have disparity > 300, as explained in [24]. Inference memory footprints are our theoretical estimates based on network structures and do not include memory required for storing networks' parameters (real memory footprint will depend on implementation). Error rates and numbers of parameters are taken from the respective publications.

| Method | Params [M] | Memory [GB] | 3EP [%] | MAE [px] | Modify. Disp. |
|---|---|---|---|---|---|
| PDS (proposed) | 2.2 | 0.4 | 3.38 (2.89) | 1.12 (0.87) | ✓ |
| PSM [2] | 5.2 | 0.6 | n/a | 1.09 | ✗ |
| CRL [24] | 78 | 0.2 | 6.20 | 1.32 | ✗ |
| iResNet-i2 [18] | 43 | 0.2 | 4.57 | 1.40 | ✗ |
| DispNetCorr1D [20] | 42 | 0.1 | n/a | 1.68 | ✗ |
| LRCR [12] | 30 | 9.0 | 8.67 | 2.02 | ✗ |
| GC [13] | 3.5 | 4.5 | 9.34 | 2.02 | ✗ |

**End-to-end deep stereo**. Recent works attempt at solving stereo matching using neural network trained end-to-end without post-processing [4, 20, 13, 43, 24, 12, 18, 2]. Such a network is typically a pipeline composed of **embedding**, **matching**, **regularization** and **refinement** modules:

The **embedding** module produces image descriptors for left and right images, and the (non-parametric) **matching** module performs an explicit correlation between shifted descriptors to compute a cost volume for every disparity [4, 20, 24, 12, 18]. This matching module may be absent, and concatenated left-right descriptors directly fed to the **regularization** module [13, 2, 43]. This strategy uses more context, but the deep network implementing such a module has a larger memory footprint as shown in Table 1. In this work we reduce memory use without sacrificing accuracy by introducing a matching module that compresses concatenated left-right image descriptors into compact matching signatures.

The **regularization** module takes the cost volume, or the concatenation of descriptors, regularizes it, and outputs either disparities [20, 4, 24, 18] or a distribution over disparities [13, 43, 12, 2]. In the latter case, sub-pixel disparities can be computed as a weighted average with SoftArgmin, which is sensitive to erroneous minor modes in the inferred distribution.

This **regularization** module is usually implemented as a hourglass deep network with shortcut connections between the contracting and the expanding parts [20, 4, 24, 13, 43, 2, 18]. It is composed of 2D convolutions and does not treat all disparities symmetrically in some models [20, 4, 24, 18], which makes the network over-parametrized and prohibits the change of the disparity range without modification of its structure and re-training. Or it can use 3D convolutions that treat all disparities symmetrically [13, 43, 12, 2]. As a consequence these networks have less parameters, but their disparity range is still non-adjustable without re-training due to SoftArgmin as we show in § 3.3. In this work, we propose to use a novel sup-pixel MAP approximation for inference which computes a mean around the disparity with minimum matching cost. It is more robust to erroneous modes in the distribution and allows to modify the disparity range without re-training.

Finally, some methods [24, 18, 12] also have a **refinement** module, that refines the initial low-resolution disparity relying on attention map, computed as left-right warping error. The training of end-to-end networks is usually performed in fully supervised manner (except of [43]).

All described methods [4, 20, 13, 43, 24, 12, 18, 2] use modest-size image patches during training. In this work, we show that training on full-size images boosts networks ability to utilize large context and improves its accuracy. Also, the methods, even the ones producing disparity distribution, rely on $L^1$ loss, since it allows to train the network to produce sub-pixel disparities. We, instead propose to use more "natural" sub-pixel cross-entropy loss that ensures faster converges and better accuracy.

**Our contributions** can be summarize as follows:

1. We decrease the memory footprint by introducing a novel bottleneck **matching** module. It compresses the concatenated left-right image descriptors into compact matching signatures, which are then concatenated and fed to the hourglass network we use as **regularization** module, instead of the concatenated descriptors themselves as in [13, 2]. Reduced memory footprint allows to process larger images and to train on full-size images, that boosts networks ability to utilize large context.

2. Instead of computing the posterior mean of the disparity and training with a $L_1$ penalty [2, 12, 43, 13] we propose for inference a sub-pixel MAP approximation that computes a expectation around the disparity with minimum matching cost, which is robust to erroneous modes in the disparity distribution and allows to modify the disparity range without re-training. For training we similarly introduce a sub-pixel criterion by combining the standard cross-entropy with a kernel interpolation, which provides faster convergence rates and higher accuracy.

In the experimental section, we validate our contributions. In § 3.2 we show how the reduced memory footprint allows to train on full-size images and to leverage large image contexts to improve performance. In § 3.3 we demonstrate that, thanks to the proposed sub-pixel MAP and cross-entropy, we are able to modify the disparity range without re-training, and to improve the matching accuracy. Then, in § 3.4 we compare our method to state-of-the-art baselines and show that it has smallest 3-pixels error (3PE) and smallest or second smallest mean absolute error (MAE) on the FlyingThings3D set, depending on the evaluation protocol and ranked third and fourth on KITTI'15 and KITTI'12 sets respectively.

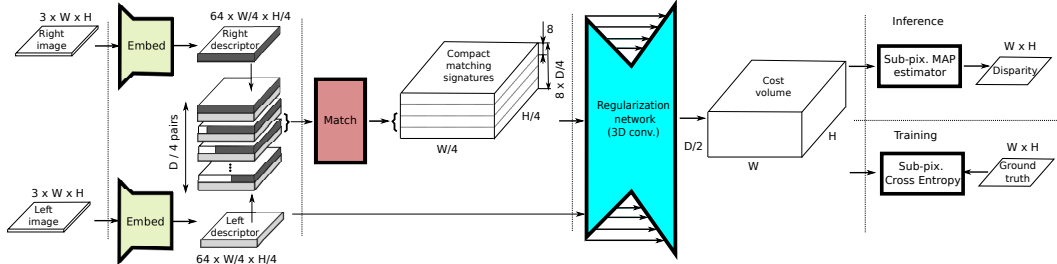

Figure 1: Network structure and processing flow during training and inference. Input / output quantities are outlined with thin lines, while processing modules are drawn with thick ones. Following the vocabulary introduced in § 1, the yellow shapes are **embedding** modules, the red rectangle the **matching** module and the blue shape the the **regularization** module. The **matching** module is a contribution of our work, as in previous methods [13, 2] left and shifted right descriptors are directly fed to the **regularization** module (hourglass network). Note that the concatenated compact matching signature tensor is a 4D tensor represented here as 3D by combining the feature indexes and disparities along the vertical axis.

## 2   Method

### 2.1   Network structure

Our network takes as input the left and right color images $\{\mathbf{x}^L, \mathbf{x}^R\}$ of size $W \times H$ and produces a "cost tensor" $\mathbf{C} = Net(\mathbf{x}^L, \mathbf{x}^R \mid \mathbf{\Theta}, D)$ of size $\frac{D}{2} \times W \times H$, where $\mathbf{\Theta}$ are the model's parameters, an $D \in \mathbb{N}$ is the maximum disparity.

The computed cost tensor is such that $\mathbf{C}_{k,i,j}$ is the cost of matching the pixel $\mathbf{x}_{i,j}^L$ in the left image to the pixel $\mathbf{x}_{i-2k,j}^R$ in the right image, which is equivalent to assigning the disparity $\mathbf{d}_{i,j} = 2k$ to the left image pixel.

This cost tensor $\mathbf{C}$ can then be converted into an a posterior probability tensor as

$$P\left(\mathbf{d} \mid \mathbf{x}^L, \mathbf{x}^R\right) = \operatorname*{softmax}_{k}\left(-\mathbf{C}_{k,i,j}\right).$$

The overall structure of the network and processing flow during training and inference are shown in Figure 1, and we can summarize for clarity the input/output to and from each of the modules:

- The **embedding** module takes as input a color image $3 \times W \times H$, and computes an image descriptor $64 \times \frac{W}{4} \times \frac{H}{4}$.

- The **matching** module takes as input, for each disparity $d$, a left and a (shifted) right image descriptor both $64 \times \frac{W}{4} \times \frac{H}{4}$, and computes a compact matching signature $8 \times \frac{W}{4} \times \frac{H}{4}$. This module is unique to our network and described in details in § 2.2.

- The **regularization** module is a hourglass 3D convolution neural network with shortcut connections between the contracting and the expanding parts. It takes a tensor composed of concatenated compact matching signatures for all disparities of size $8 \times \frac{D}{4} \times \frac{W}{4} \times \frac{H}{4}$, and computes a matching cost tensor $\mathbf{C}$ of size $\frac{D}{2} \times W \times H$.

Additional information such as convolution filter size or channel numbers is provided in the Supplementary materials.

According to the taxonomy in [28] all traditional stereo matching methods consist of (1) matching cost computation, (2) cost aggregation, (3) optimization, and (4) disparity refinement steps. In the proposed network, the **embedding** and the **matching** modules are roughly responsible for the step (1) and the **regularization** module for the steps (2-4).

Besides the **matching** module, there are several other design choices that reduce test and training memory footprint of our network. In contrast to [13] we use aggressive four-times sub-sampling in the **embedding** module, and the hourglass DNN we use for **regularization** module produces probabilities only for even disparities. Also, after each convolution and transposed convolution in our network we place Instance Normalization (IN) [35] instead of Batch Normalization (BN), since we use individual full-size images during training.

## 2.2 Matching module

The core of state-of-the-art methods [13, 43, 12, 2] is the 3D convolutions Hourglass network used as **regularization** module, that takes as input a tensor composed of concatenated left-right image descriptor for all possible disparity values. The size of this tensor makes such networks have a huge memory footprint during inference.

We decrease the memory usage by implementing a novel **matching** with a DNN with a "bottleneck" architecture. This module compresses the concatenated left-right image descriptors into a compact matching signature for each disparity, and the results is then concatenated and fed to the Hourglass module. This contrasts with existing methods, which directly feed the concatenated descriptors [13, 43, 12, 2] to the Hourglass regularization module. For example, while in [13] authors feed 64 channels 3D tensor to the regularization network, we feed 8 channels tensor and reach a similar accuracy. Reducing the memory footprint allows to process a larger area during inference, and consequently to use a larger context to estimate disparity which solve ambiguities and translates directly into better performance.

This module is inspired by CRL [24] and DispNetCorr1D [24, 20] which control the memory footprint (as shown in Table 1) by feeding correlation results instead of concatenated embeddings to the Hourglass network and by [38] that show superior performance of joint left-right image embedding. We also borrowed some ideas from the bottleneck module in ResNet [10], since it also encourages compressed intermediate representations.

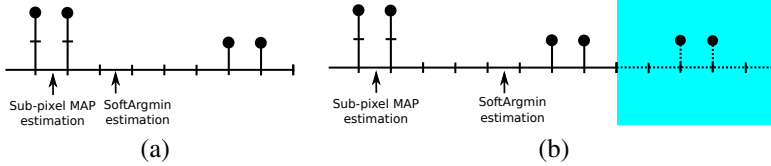
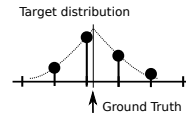

Figure 2: Comparison of the proposed Sub-pixel MAP with the standard SoftArgmin: (a) in presence of a multi-modal distribution SoftArgmin blends all the modes and produces an incorrect disparity estimate. (b) when the disparity range is extended (blue area), SoftArgmin estimate may degrade due to additional modes.

Figure 3: Target distribution of sub-pixel cross-entropy is a discretized Laplace distribution centered at sub-pixel ground-truth disparity.

## 2.3 Sub-pixel MAP

In state-of-the-art methods, a network produces an posterior disparity distribution and then use a SoftArgmin module [13, 43, 12, 2], introduced in [13], to compute the predicted sub-pixel disparity as an expectation of this distribution[1]:

$$\hat{d} = \sum_d d \cdot P\left(\mathbf{d} = d \mid \mathbf{x}^L, \mathbf{x}^R\right)$$

This SoftArgmin approximates a sub-pixel maximum a posteriori (MAP) solution when the distribution is unimodal and symmetric. However, as illustrated in Figure 2, this strategy suffers from two key weaknesses: First, when these assumptions are not fulfilled, for instance if the posterior is multi-modal, this averaging blends the modes and produces a disparity estimate far from all of them. Second, if we want to apply the model to a greater disparity range without re-training, the estimate may degrade even more due to additional modes.

The authors of [13] argue that when the network is trained with the SoftArgmin, it adapts to it during learning by rescaling its output values to make the distribution unimodal. However, the network learns rescaling only for disparity range used during training. If we decide to change the disparity range during the test, we will have to re-train the network.

To address both of these drawbacks, we propose to use for inference a sub-pixel MAP approximation that computes a expectation around the disparity with minimum matching cost as

$$\tilde{d} = \sum_d d \cdot P\left(\mathbf{d} = d \mid \mathbf{x}^L, \mathbf{x}^R\right), \text{ where}$$

$$P\left(\mathbf{d} \mid \mathbf{x}^L, \mathbf{x}^R\right) = \operatorname*{softmax}_{d:\left|\hat{d}-d\right| \leq \delta}\left(-\mathbf{C}_{d,x,y}\right) \text{ and } \hat{d} = \operatorname*{arg\,min}_d\left(\mathbf{C}_{d,x,y}\right) \quad (1)$$

with $\delta$ a meta-parameter (in our experiments we choose $\delta = 4$ based on small scale grid search experiment on the validation set). The approximation works under assumption that the distribution is symmetric in a vicinity of a major mode.

In contrast to the SoftArgmin, the proposed sup-pixel MAP is used only for inference. During training we use the posterior disparity distribution and the sub-pixel cross-entropy loss discussed in the next section.

## 2.4 Sub-pixel cross-entropy

Many methods use the $L^1$ loss [2, 12, 43, 13], even though the "natural" choice for the classification by design networks, producing distribution over discrete disparity values is a cross-entropy. The $L^1$ loss is often selected because it empirically [13] performs better than cross-entropy, and because when it is combined with SoftArgmin, it allows to train a network with sub-pixel ground truth.

In this work, we propose a novel sub-pixel cross-entropy that provides faster convergence and better accuracy. The target distribution of our cross-entropy loss is a discretized Laplace distribution centered at the ground-truth disparity $d^{gt}$, shown in Figure 3 and computed as

$$Q^{gt}(d) = \frac{1}{N} \exp\left(-\frac{|d - d^{gt}|}{b}\right), \quad \text{where} \quad N = \sum_i \exp\left(-\frac{|i - d^{gt}|}{b}\right),$$

where $b$ is a diversity of the Laplace distribution (in our experiments we set $b = 2$, reasoning that the distribution should cover at least several discrete disparities). With this target distribution we compute cross-entropy as usual

$$L(\mathbf{\Theta}) = \sum_d Q^{gt}(d) \cdot \log P\left(\mathbf{d} = d \mid \mathbf{x}^L, \mathbf{x}^R, \mathbf{\Theta}\right). \tag{2}$$

The proposed sub-pixel cross-entropy is different from soft cross entropy [19], since in our case probability in each discrete location of the target distribution is a smooth function of a distance to the sub-pixel ground-truth. This allows to train the network to produce a distribution from which we can compute sub-pixel disparities using our sub-pixel MAP.

## 3   Experiments

Our experiments are done with the PyTorch framework [26]. We initialize weights and biases of the network using default PyTorch initialization and train the network as shown in Table 2. During the training we normalize training patches to zero mean and unit variance. The optimization is performed with the RMSprop method with standard settings.

Table 2: Summary of training settings for every dataset.

|                | FlyingThings3D | KITTI |
|----------------|----------------|-------|
| Mode           | from scratch   | fine-tune |
| Lr. schedule   | 0.01 for 120k, half every 20k | 0.005 for 50k, half every 20k |
| Iter. #        | 160k           | 100k  |
| Tr. image size | $960 \times 540$ full-size | $1164 \times 330$ |
| Max disparity  | 255            | 255   |
| Augmentation   | not used       | mixUp [42], anisotropic zoom, random crop |

We guarantee reproducibility of all experiments in this section by using only available data-sets, and making our code available online under open-source license after publication.

### 3.1   Datasets and performance measures

We used three data-sets for our experiments: KITTI'12 [6] and KITTI'15 [22], that we combined into a KITTI set, and FlyingThings3D [20] summarized in Table 3. KITTI'12, KITTI'15 sets have online scoreboards [15].

The FlyingThings3D set suffers from two problems: (1) as noticed in [24, 42], some images have very large (up to $10^3$) or negative disparities; (2) some images are rendered with black dots artifacts. For the training we use only images without artifacts and with disparities $\in [0, 255]$.

To deal with this problem, in some previous publications authors process the test set *using the ground truth* which is used for benchmarking. Such pre-processing may consist of ignoring pixels with disparity $> 192$ [2], or discarding images with more than $25\%$ of pixels with disparity $> 300$ [24, 18, 20]. For the sake of fairness of the comparison we computed the error using both protocols during the benchmarking on FlyingThings3D set. In all other experiments we use the unaltered test set.

We make validation sets by withholding 500 images from the FlyingThings3D training set, and 58 from the KITTI training set, respectively.

We measure the performance of the network using two standard measures: (1) *3-pixel-error (3PE)*, which is the percentage of pixels for which the predicted disparity is off by more than 3 pixels, and

Table 3: Datasets used for experiments. During benchmarking, we follow previous works and use maximum disparity, that is different from absolute maximum for the datasets, provided between parentheses.

| Dataset | Test # | Train # | Size | Max disp. | Ground truth | Web score |
|---|---|---|---|---|---|---|
| KITTI | 395 | 395 | $1226 \times 370$ | 192 (230) | sparse, $\leq 3$ px. | ✓ |
| FlyingThings3D | 4370 | 25756 | $960 \times 540$ | 192 (6773) | dense , unknown | ✗ |

Table 4: Error of the proposed PDS network on FlyingThings3d test set as a function of training patch size. The network trained on full-size images (highlighted), outperforms the network trained on small image patches. Note, that in this experiment we used SoftArgmin with $L^1$ loss during training.

| Train size | Test size | 3PE, [%] | MAE, [px] |
|---|---|---|---|
| $512 \times 256$ | $512 \times 256$ | 8.63 | 4.18 |
| $512 \times 256$ | $960 \times 540$ | 5.28 | 3.55 |
| $960 \times 540$ | $960 \times 540$ | 4.50 | 3.40 |

(2) *mean-absolute-error (MAE)*, the average difference of the predicted disparity and the ground truth. Note, that 3PE and MAE are complimentary, since 3PE characterize error robust to outliers, while MAE accounts for sub-pixel error.

## 3.2 Training on full-size images

In this section we show the effectiveness of training on full-size images. For that we train our network till convergence on FlyingThings3D dataset with the $L^1$ loss and SoftArgmin twice, the first time we use $512 \times 256$ training patches randomly cropped from the training images as in [13, 2], and the second time we used full-size $960 \times 540$ training images. Note, that the latter is possible thanks to the small memory footprint of our network.

As seen in Table 4, the network trained on small patches, performs better on larger than on smaller test images. This suggests, that even the network that has not seen full-size images during training can utilize a larger context. As expected, the network trained on full-size images makes better use of the said context, and performs significantly better.

## 3.3 Sub-pixel MAP and cross-entropy

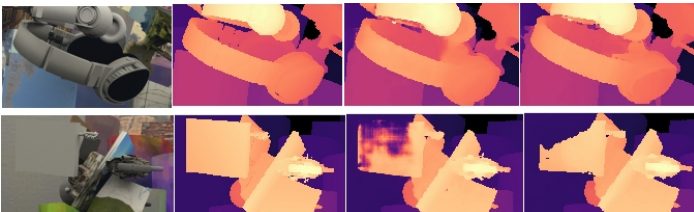 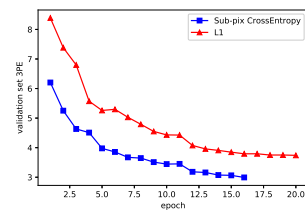

Figure 4: Example of disparity estimation errors with the SoftArgmin and sup-pixel MAP on FlyingThings3d set. The first column shows the input image, the second – ground truth disparity, the third – SoftArgmin estimate and the fourth sub-pixel MAP estimate. Note that SoftArgmin estimate, though wrong, is closer to the ground truth than sub-pixel MAP estimate. This can explain larger MAE of the sub-pixel MAP estimate.

Figure 5: Comparison of the convergence speed on FlyingThings3d set with sub-pixel cross entropy and $L^1$ loss. With the proposed sub-pixel cross-entropy loss (blue) network converges faster. Note, that the error is computed using the validation set, containing 500 examples.

In this section, we first show the advantages of the sub-pixel MAP over the SoftArgmin. We train our PDS network till convergence on FlyingThings3D with SoftArgmin, $L^1$ loss and full-size training

images and then test it twice: the first time with SoftArgmin for inference, and the second time with our sub-pixel MAP for inference instead.

As shown in Table 5, the substitution leads to the reduction of the 3PE and slight increase of the MAE. The latter probably happens because in the erroneous area SoftArgmin estimates are wrong, but nevertheless closer to the ground truth since it blends all distribution modes, as shown in Figure 4.

Table 5: Performance of the sub-pixel MAP estimator and cross-entropy loss on FlyingThings3d set. Note, that: (1) if we substitute SoftArgmin with sub-pixel MAP during the test we get lower 3PE and similar MAE; (2) if we increase disparity range twice MAE and 3PE of the network with sub-pixel MAP almost does not change, while errors of the network with SoftArgmin increase; (3) if we train network with with sub-pixel cross entropy it has much lower 3PE and only slightly worse MAE.

| Loss | Estimator | 3PE, [%] | MAE, [px] |
|---|---|---|---|
| **Standard disparity range** $\in [0, 255]$ | | | |
| $L^1$ + SoftArgmin | SoftArgmin | 4.50 | 3.40 |
| $L^1$ + SoftArgmin | Sub-pixel MAP | 4.22 | 3.42 |
| Sub-pixel cross-entropy. | Sub-pixel MAP | 3.80 | 3.63 |
| **Increased disparity range** $\in [0, 511]$ | | | |
| $L^1$ + SoftArgmin | SoftArgmin | 5.20 | 3.81 |
| $L^1$ + SoftArgmin | Sub-pixel MAP | 4.27 | 3.53 |

When we test the same network with the disparity range increased from 255 to 511 pixels the performance of the network with the SoftArgmin plummets, while performance of the network with sub-pixel MAP remains almost the same as shown in Table 5. This shows that with Sub-pixel MAP we can modify the disparity range of the network on-the-fly, without re-training.

Next, we train the network with the sub-pixel cross-entropy loss and compare it to the network trained with SoftArgmin and the $L^1$ loss. As show in Table 5, the former network has much smaller 3PE and only slightly larger MAE. The convergence speed with sub-pixel cross-entropy is also much faster than with $L^1$ loss as shown in Figure 5. Interestingly, in [13] also reports faster convergence with one-hot cross-entropy than with $L^1$ loss, but contrary to our results, they found that $L^1$ provided smaller 3PE.

### 3.4 Benchmarking

In this section we show the effectiveness of our method, compared to the state-of-the-art methods. For KITTI, we computed disparity maps for the test sets with withheld ground truth, and uploaded the results to the evaluation web site. For the FlyingThings3D set we evaluated performance on the test set ourselves, following the protocol of [2] as explained in § 3.1.

**FlyingThings3D set** benchmarking results are shown in Table 1. Notably, the method we propose has lowest 3PE error according to both evaluation protocols and has lowest or second lowest MAE, depending on the protocol. Moreover, in contrast to other methods, our method has small memory footprint, number of parameters, and it allows to change the disparity range without re-training.

**KITTI'12, KITTI'15** benchmarking results are shown in Table 6. The method we propose ranks third on KITTI'15 set and fourth on KITTI'12 set, taking into account state-of-the-art results published a few months ago or not officially published yet iResNet-i2 [18], PSMNet [2] and LRCR [12] methods.

## 4   Conclusion

In this work we addressed two issues precluding the use of deep networks for stereo matching in many practical situations in spite of their excellent accuracy: their large memory footprint, and the inability to adjust to a different disparity range without complete re-training.

We showed that by carefully revising conventionally used networks architecture to control the memory footprint and adapt analytically the network to the disparity range, and by using a new loss and estimator to cope with multi-modal posterior and sub-pixel accuracy, it is possible to resolve these practical issues and reach state-of-the-art performance.

Table 6: KITTI'15 (top) and KITTI'12 (bottom) snapshots from 15/05/2018 with top-10 methods, including published in a recent months on not officially published yet: iResNet-i2 [18], PSMNet [2] and LRCR [12]. Our method (highlighted) is 3rd in KITTI'15 and 4th in KITTI'12 leader boards.

| # | dd/mm/yy | Method | 3PE (all pixels), [%] | Time, [s] |
|---|----------|--------|------------------------|-----------|
| 1 | 30/12/17 | PSMNet [2] | 2.16 | 0.4 |
| 2 | 18/03/18 | iResNet-i2 [18] | 2.44 | 0.12 |
| 3 | 15/05/18 | PDS (proposed) | 2.58 | 0.5 |
| 4 | 24/03/17 | CRL [24] | 2.67 | 0.47 |
| 5 | 27/01/17 | GC-NET [13] | 2.87 | 0.9 |
| 6 | 15/11/17 | LRCR [12] | 3.03 | 49 |
| 7 | 15/11/16 | DRR [7] | 3.16 | 0.4 |
| 8 | 08/11/17 | SsSMnet [43] | 3.40 | 0.8 |
| 9 | 15/12/16 | L-ResMatch [31] | 3.42 | 48 |
| 10 | 26/10/15 | Displets v2 [8] | 3.43 | 265 |

| # | dd/mm/yy | Method | 3PE (non-occluded), [%] | Time, [s] |
|---|----------|--------|--------------------------|-----------|
| 1 | 31/12/17 | PSMNet [2] | 1.49 | 0.4 |
| 2 | 23/11/17 | iResNet-i2 [18] | 1.71 | 0.12 |
| 3 | 27/01/17 | GC-NET [13] | 1.77 | 0.9 |
| 4 | 15/05/18 | PDS (proposed) | 1.92 | 0.5 |
| 5 | 15/12/16 | L-ResMatch [31] | 2.27 | 48 |
| 6 | 11/09/16 | CNNF+SGM [41] | 2.28 | 71 |
| 7 | 15/12/16 | SGM-NET [30] | 2.29 | 67 |
| 8 | 08/11/17 | SsSMnet [43] | 2.30 | 0.8 |
| 9 | 27/04/16 | PBCP [29] | 2.36 | 68 |
| 10 | 26/10/15 | Displets v2 [8] | 2.37 | 265 |

## 5  Acknowledgement

We gratefully acknowledge support from the NCCR PlanetS and CaSSIS project of the University of Bern funded through the Swiss Space Office via ESA's PRODEX program. We also acknowledge the support of NVIDIA Corporation with the donation of the GeForce GTX TITAN X used for this research.

## Footnotes

[1]The name SoftArgmin comes from the fact that the function computes disparity of the match with the minimum matching cost in a "soft" way. The matching cost, unlike likelihood probability, is small for correct matches and large for incorrect ones. However, it can be also interpreted as expectation of probability distribution over disparities.

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
