[Supplementary Material · supplementary.pdf]

Table 1: Structure of networks' modules. Each Residual block consists of two 2D convolutions followed by shortcut connection. Every convolution and transposed convolution (including these in the Residual blocks), is followed by LeakyReLU with negative slope 0.2 and Instance Normalization.

| | Layer Description | Output Dimension |
|---|---|---|
| | color image | $H \times W \times 3$ |
| **Embedding module** | | |
| E1 | 2D conv. $3 \times 5 \times 5 \times 64$ stride 2 | $\frac{1}{2}W \times \frac{1}{2}W \times 64$ |
| E2 | 2D conv. $64 \times 5 \times 5 \times 64$ stride 2 | $\frac{1}{4}H \times \frac{1}{4}W \times 64$ |
| E3 | $2\times$ Residual block with $64 \times 3 \times 3 \times 64$ 2D conv. | $\frac{1}{4}H \times \frac{1}{4}W \times 64$ |
| E4-redir. | 2D conv. $64 \times 3 \times 3 \times 8$ no IN, LeakyReLU | $\frac{1}{4}H \times \frac{1}{4}W \times 8$ |
| **Matching module** | | |
| M1 | concatenate left-right embeddings E3 | $\frac{1}{4}H \times \frac{1}{4}W \times 128$ |
| M2 | 2D conv. $128 \times 3 \times 3 \times 64$ | $\frac{1}{4}H \times \frac{1}{4}W \times 64$ |
| M3 | $2\times$ Residual block with $64 \times 3 \times 3 \times 64$ 2D conv. | $\frac{1}{4}H \times \frac{1}{4}W \times 64$ |
| M4 | 2D conv. $64 \times 3 \times 3 \times 8$ no IN, LeakyReLU | $\frac{1}{4}H \times \frac{1}{4}W \times 8$ |
| **Regularization module** | | |
| H1 | concatenate joint embeddings M4 | $\frac{1}{4}H \times \frac{1}{4}W \times \frac{1}{4}D \times 8$ |
| H2 | 3D conv. $8 \times 3 \times 3 \times 3 \times 8$ | $\frac{1}{4}H \times \frac{1}{4}W \times \frac{1}{4}D \times 8$ |
| H3 | 3D conv. $8 \times 3 \times 3 \times 3 \times 16$, stride 2 | $\frac{1}{8}H \times \frac{1}{8}W \times \frac{1}{8}D \times 16$ |
| H4 | H3 + E4-redir. | $\frac{1}{8}H \times \frac{1}{8}W \times \frac{1}{8}D \times 16$ |
| H5 | 3D conv. $16 \times 3 \times 3 \times 3 \times 16$ | $\frac{1}{8}H \times \frac{1}{8}W \times \frac{1}{8}D \times 16$ |
| H6 | H5 + H4 | $\frac{1}{8}H \times \frac{1}{8}W \times \frac{1}{8}D \times 16$ |
| H7 | 3D conv. $16 \times 3 \times 3 \times 3 \times 32$, stride 2 | $\frac{1}{16}H \times \frac{1}{16}W \times \frac{1}{16}D \times 32$ |
| H8 | 3D conv. $32 \times 3 \times 3 \times 3 \times 32$ | $\frac{1}{16}H \times \frac{1}{16}W \times \frac{1}{16}D \times 32$ |
| H9 | H8 + H7 | $\frac{1}{16}H \times \frac{1}{16}W \times \frac{1}{16}D \times 32$ |
| H10 | 3D conv. $32 \times 3 \times 3 \times 3 \times 64$, stride 2 | $\frac{1}{32}H \times \frac{1}{32}W \times \frac{1}{32}D \times 64$ |
| H11 | 3D conv. $64 \times 3 \times 3 \times 3 \times 64$ | $\frac{1}{32}H \times \frac{1}{32}W \times \frac{1}{32}D \times 64$ |
| H12 | H11 + H10 | $\frac{1}{32}H \times \frac{1}{32}W \times \frac{1}{32}D \times 64$ |
| H13 | 3D conv. $64 \times 3 \times 3 \times 3 \times 128$, stride 2 | $\frac{1}{64}H \times \frac{1}{64}W \times \frac{1}{64}D \times 128$ |
| H14 | 3D dconv. $128 \times 4 \times 4 \times 4 \times 64$, stride 2 | $\frac{1}{32}H \times \frac{1}{32}W \times \frac{1}{32}D \times 64$ |
| H15 | H14+H11 | $\frac{1}{32}H \times \frac{1}{32}W \times \frac{1}{32}D \times 64$ |
| H16 | 3D conv. $64 \times 3 \times 3 \times 3 \times 64$ | $\frac{1}{32}H \times \frac{1}{32}W \times \frac{1}{32}D \times 64$ |
| H17 | 3D dconv. $64 \times 4 \times 4 \times 4 \times 32$, stride 2 | $\frac{1}{16}H \times \frac{1}{16}W \times \frac{1}{16}D \times 32$ |
| H18 | H17+H8 | $\frac{1}{16}H \times \frac{1}{16}W \times \frac{1}{16}D \times 32$ |
| H19 | 3D conv. $32 \times 3 \times 3 \times 3 \times 32$ | $\frac{1}{16}H \times \frac{1}{16}W \times \frac{1}{16}D \times 32$ |
| H20 | 3D dconv. $32 \times 4 \times 4 \times 4 \times 16$, stride 2 | $\frac{1}{8}H \times \frac{1}{8}W \times \frac{1}{8}D \times 16$ |
| H21 | H20+H5 | $\frac{1}{8}H \times \frac{1}{8}W \times \frac{1}{8}D \times 16$ |
| H22 | 3D conv. $16 \times 3 \times 3 \times 3 \times 16$ | $\frac{1}{8}H \times \frac{1}{8}W \times \frac{1}{8}D \times 16$ |
| H23 | 3D dconv. $16 \times 4 \times 4 \times 4 \times 8$, stride 2 | $\frac{1}{4}H \times \frac{1}{4}W \times \frac{1}{4}D \times 8$ |
| H24 | H23+H3 | $\frac{1}{4}H \times \frac{1}{4}W \times \frac{1}{4}D \times 8$ |
| H25 | 3D conv. $8 \times 3 \times 3 \times 3 \times 8$ | $\frac{1}{4}H \times \frac{1}{4}W \times \frac{1}{4}D \times 8$ |
| H26 | 3D dconv. $8 \times 4 \times 4 \times 4 \times 4$, stride 2 | $\frac{1}{2}H \times \frac{1}{2}W \times \frac{1}{2}D \times 4$ |
| H27 | 3D dconv. $4 \times 3 \times 4 \times 4 \times 1$, stride (1,2,2) no IN, LeakyReLU | $H \times W \times \frac{1}{2}D$ |