[Reviews · NeurIPS 2018]

Reviewer 1



Post-rebuttal: I thank the authors for the informative rebuttal and apologize for overlooking some details. Considering the rebuttal, I think the paper is rather on the accept side. --- The paper proposes two modifications to end-to-end disparity estimation pipelines: bottleneck matching module and “sub-pixel MAP” approach to computing the disparity from a predicted distribution over disparities. The method is evaluated on FlyingThings3D and KITTI’12 and ’15 datasets. The first modification reduces the computational requirements (in particular, memory). The second one improves the performance. Pros: 1) The modifications are simple and seem efficient. I have not seen these in previous works, although I must admit I am not intimately familiar with the details of loss functions of all recent disparity estimation works. 2) The second contribution (“sub-pixel MAP”) is evaluated quite thoroughly and provides improvement relative to a simpler SoftArgmin-based alternative. 3) The method’s performance on FlyingThings3D is significantly better than prior published work, on KITTI - roughly on par with prior published work. 4) The paper is written quite clearly. Cons: 1) Basically, the key contribution of the paper is “sub-pixel MAP” and “sub-pixel cross-entropy” loss. The first is smoothed argmax. The second is cross-entropy loss where the target delta-function is smoothed with a Laplacian. These are reasonable engineering decisions, but they do not necessarily offer much generally relevant scientific insight. 2) The experimental results are good, but on KITTI the method is not clearly outperforming existing methods, for instance it’s close to GC-NET (ICCV 2017). On FlyingThings3D the advantage is much more visible, but FlyingThings3D is not a real-world dataset, so the results there have somewhat smaller weight. 3) Some aspects of the experiments are confusing: a) lines 181-186 mention ignoring large disparities in evaluation and argue that this is made to be compatible with PSM. However, PSM was not officially published at the time of submission, so does not serve as an actual baseline: actual baselines are prior published methods. The comparison should be fair with respect to those, but is it? If not, this is a problem. Ideally, both numbers can be reported: ignoring or not ignoring large disparities. b) What is the relation between Figure 5 and Table 5? In Figure 5 Sub-pixel cross-entropy is much better than L1 in terms of 3PE, reaching roughly a value of 3. In Table 5 it has 3.80 3PE and the difference with L1 seems somewhat smaller. Why is this? 4) The “bottleneck matching module” is almost not discussed and not evaluated. How much benefit exactly does it bring? Does it affect the performance? It is nice that it allows training with larger images, but more details are needed. 5) Writing is sloppy at times (proofreading would help). In particular: a) It seems there is some confusion between argmin and argmax in the method description: the “baseline” approach is referred to as SoftArgmin, while the illustration and the formulas for the proposed method show argmax. This is probably because of some slight variations in the formulation, but would be nice to make this consistent. b) Training details are missing, such as the used optimizer, the learning rate, the duration of training, etc. This makes the results unreproducible. To conclude, the paper describes two improvements to end-to-end disparity estimation systems which allow to improve the performance and reduce the computational load. The improvements seem to work well, but they are quite minor and incremental, so it is unclear if they will be of interest to the NIPS audience. I am in a somewhat borderline mode.

Reviewer 2



The paper proposes a new deep learning-based stereo algorithm that achieves high accuracy, is memory and compute efficient, and allows their network to be applied to be adapted to different disparity ranges without re-training. The latter two properties are aimed towards increasing the ease of practical use in real-world stereo systems. The main contributions of the paper are in the architecture design (the structure of the matching modules, and how their outputs are then fed to a regularization network) that seeks to minimize memory usage while still allowing the expression of computational steps typically known to be useful for stereo---indeed, the paper discusses how the different components of their network relate to a traditional taxonomy of stereo processing steps. The second contribution is in treating the final output as a 'classification' (where a regression would fix the disparity range), with a modified averaging scheme around the mode of the output to achieve sub-pixel accuracy. The paper credits this with endowing their method to succeed at different disparity ranges without retraining. The paper is well written and motivated, and achieves good experimental performance. I believe that since [2] and [18] were officially published after the NIPS submission deadline, they should be considered as concurrent submissions (even though I appreciate the authors including these works in the final table). Given this, the paper achieves state of the art performance while being computationally efficient and maintaining a relatively low footprint. Post Rebuttal After reading the other reviews and rebuttal, I do believe this paper to be above the accept threshold. I believe there is sufficient value in the authors' contribution of coming up with a computationally efficient yet accurate architecture for stereo / correspondence problems, and goes a long way in making deep-learning based stereo more useful in practical settings.

Reviewer 3



Summary ------- This paper presents a method for computing a disparity map from a rectified pair of images. The authors propose an end-to-end learning approach consisting of three modules: the embedding module, the matching module, and the regularization module. The embedding module extracts feature vectors from the image par. The feature vectors from the left image and right image are horizontally offset and concatenated (this is repeated for all disparities under consideration) and the matching module is applied to compress the concatenated feature vectors. The resulting volume is processed by the regularization module which returns a distribution over disparities for each spatial location. The network is trained on full-resolution images. The target for training is a discretiezed Laplace distribution centered around the ground truth disparity and the cross-entropy loss is used to train the network. At inference the disparity is computed using sub-pixel MAP approximation, which is defined as the expected value of disparity taking into consideration only disparity values around the disparity with maximum posterior probability. The parts of the architecture that are novel are the matching module, sub-pixel MAP approximation, and using a discretized Laplace distribution as target. Strengths --------- - The authors submitted the generated disparity maps to the KITTI online evaluation server. The method performs well and is bested by only a couple of recently published papers. The KITTI leaderboard is very competitive and placing 3rd and 4th is respectable. The results on KITTI are the main reason why I leaned towards accepting the paper. - The code will be open-sourced after publication, which makes it easy to reproduce the results and build upon the ideas presented in this work. - Ablation studies on the sub-pixel MAP estimator are performed. Weaknesses ---------- - The contributions of this work (the matching module, sub-pixel MAP approximation, and using a discretized Laplace distribution during training) are solid, but not ground-breaking. Comments -------- Line 62 and line 153: The authors say that using cross-entropy is more "natural" than using L1 loss. Can you explain what you mean by natural? Line 70: "training with a vanilla L1 penalty". I'm not sure what the word "vanilla" is referring to. Maybe the authors should consider omitting it? In equation after line 91 the authors apply the softmax function to a three dimensional tensor, whereas the standard definition softmax is on vectors. My guess is that the softmax function is applied on each spatial location. Perhaps the text could mention this or the equation should be written more precisely. Line 181: "by processing the test set using the ground truth for benchmarking". I'm slightly confused by the meaning of this sentence. My interpretation was that that "they processed the test set, which is used for benchmarking". Line 182 "without mentioning it" and line 184 "although this is not commendable". I would consider omitting these two sentences, because they don't really convey any meaning and come across as slightly negative. Figure 4 and line 210: "Note that SoftArgmin estimate, though completely wrong, is closer to the ground truth than sub-pixel MAP estimate". Maybe consider omitting the work "completely" or re-editing this sentence. After all, If an estimate is completely wrong it can't be better than another estimate. Spelling -------- Line 46: "It composed of" -> "It is composed of" Line 47: "and not treat" -> "and does? not treat" Line 51: "their disparity range is still is non-adjustable" -> "their disparity range is still non-adjustable" Line 61: "it allows to train network to" -> "it allows to train the network to" Line 68: "to train on a full-size images" -> "to train on full-size images" Line 80: "Than, in 3.4 we" -> "Then, in 3.4 we" Line 205: "we firstly show" -> "we first? show" Line 206: "we train the our PDS network" -> "we train our PDS network" Line 210: "SoftArgmin estimate are" -> "SoftArgmin estimates? are" Figure 2: "Comparison the proposed" -> "Comparison of? the proposed" Figure 4: "The first column shows image" -> "The first column shows the? input? image"